# Adaptation and Validation of the Intercultural Effectiveness Scale in a Sample of Initial Teacher Training Students in Chile

**DOI:** 10.3390/bs14100864

**Published:** 2024-09-25

**Authors:** Juan Carlos Beltrán-Véliz, José Luis Gálvez-Nieto, Maura Klenner-Loebel, Nathaly Vera-Gajardo

**Affiliations:** 1Núcleo Científico Tecnológico en Ciencias Sociales y Humanidades, Universidad de La Frontera, Temuco 4811230, Chile; juan.beltran@ufrontera.cl; 2Departamento de Trabajo Social, Universidad de La Frontera, Temuco 4811230, Chile; jose.galvez@ufrontera.cl; 3Departamento de Lenguas, Literatura y Comunicación, Universidad de La Frontera, Temuco 4811230, Chile; 4Facultad de Educación, Universidad Autónoma de Chile, Temuco 4810101, Chile; nathaly.vera@uautonoma.cl

**Keywords:** intercultural effectiveness, students, teachers, validity, reliability

## Abstract

Intercultural effectiveness is a relevant construct for improving the training of future teachers and promoting culturally diverse educational environments. This study aimed to adapt and validate the intercultural effectiveness scale (IES) in a sample of pre-service teachers in Chile. A cross-sectional design study was conducted, in which 584 Chilean university students participated (male = 37.8%; female = 61.6%; other = 0.5%), with a mean age of 20.56 years (SD = 3.37) and 21.9% identifying themselves as belonging to an ethnic group. The results obtained from structural equation modelling confirmed the structure of six correlated factors. The scores of the six factors of the IES presented positive and statistically significant correlations with the intercultural sensitivity scale (ISS). In addition, the factors presented adequate levels of reliability. The results of this research are discussed based on current theoretical and empirical evidence.

## 1. Introduction

In recent years, the migration of people worldwide has increased considerably [1,2,3], and the Chilean context is no exception. Currently, Chile is a culturally diverse country due to the presence of people belonging to native cultures, which exceeds 10% [4], and the significant increase in immigrants settled in the country [5,6]. Different educational and sociocultural tensions and challenges have arisen in the social sphere [7,8], resulting in inequalities, stigmatization, racism, and marginalization [2,9,10,11]. These tensions are partly due to the difficulties that arise in the communication process between people from diverse cultures, which is affected by each culture’s worldviews and cultural patterns, resulting in misinterpretations and misunderstandings of the messages delivered [12,13].

In this respect, teachers, and their interventions, play a relevant role as social agents that promote the development of intercultural competencies in their students, who are part of increasingly multicultural societies and who need to get along adequately with people from cultures different from their own, both in the educational context and in future work and social contexts. The development of intercultural communicative competence in teachers impacts the quality of the educational processes among students in culturally diverse contexts, given that teachers must mobilize efficient scaffolding, recognizing in each of their students their culture, personal characteristics, and previous experiences, in order to generate new knowledge, skills, abilities, and attitudes and, in turn, develop appropriate and quality management in culturally heterogeneous classrooms [14,15]. Therefore, in initial teacher training programs, intercultural communication competence (ICC) should be developed with the purpose that teachers acquire a set of skills and behaviors that allow them to communicate appropriately and effectively with people from their own culture and with people who come from other cultures [16,17,18,19,20,21]. For this, it is necessary and vital to develop intercultural effectiveness in the initial teacher training since it is the behavioral aspect of ICC.

The concept of intercultural communicative competence has been referred to as such in specialized literature since the late 1980s [22]. Although it has been approached from various perspectives and fields of human activity [23,24,25,26,27,28,29,30,31], there is still no conclusive definition of the concept. However, it seems that in the different fields in which it has been studied, it is related to the skills, behaviors, and knowledge that a person needs to be successful when interacting with others from different cultures. This instability of the concept could have its origin in the changes that global communication has undergone due to technological advances, which have broadened its scope and understanding [19].

Several theoretical models have been developed to understand ICC. Hammer [32,33], based on Bennett’s [34,35,36] developmental model of intercultural sensitivity (DMIS), offers a model called the intercultural development continuum. This model understands intercultural communicative competence in terms of five stages of development: denial (lack of awareness of the importance of the cultural aspect), polarization (surprise at the foreignness), minimization (identification of commonalities), acceptance (understanding and appreciation of belief systems), and adaptation (behavior adapted to intercultural dialogue). Hammers’ model has been reviewed and metaphorically represented as a pendulum by Acheson and Schneider-Bean [37] rather than a linear continuum. This agrees with research findings related to the cultural immersion experiences of migrant subjects. In this metaphor, the fluid and complex nature of the development and maintenance of intercultural communicative competence is clear, highlighting how this competence has globally applicable characteristics, i.e., regardless of the culture with which the subject establishes a relationship, and at the same time is dependent on specific intercultural encounters and experiences. For his part, Byram [38] proposes a model of ICC organized around six knowledge components (savoirs’): knowledge (savoir), interpretation and relationship skills (savoir comprendre), discovery and interaction skills (savoir faire), attitudes (savoir ëtre) and critical cultural awareness (savoir engager).

Chen [19] suggests that the previously described models contain certain conceptual ambiguities in the theoretical assumptions of the components of ICC, which can lead to problems when operationalizing them, i.e., when assessing the development of this competence in individuals. Therefore, the author presents a triangular model of ICC composed of three constructs: intercultural sensitivity, effectiveness, and awareness. Intercultural sensitivity (IS) is related to the affective component of ICC, i.e., the emotions originating from individuals, which in turn are elicited by other individuals in intercultural dialogue experiences and scenarios [39]. Intercultural effectiveness (IE) (also called adroitness) is the behavioral area of ICC [40], i.e., how capable individuals are of understanding and delivering messages appropriate to interaction with people from cultures different from their own. Intercultural awareness (IA) is related to the cognitive component of ICC. It encompasses knowledge of cultures and how culture affects communication, and knowledge of social conventions and how these affect the manner in which individuals think and interact [41]. Social conventions vary from culture to culture. Following this review of ICC models, Chen’s proposal of an ICC model understood in terms of constructs (IA, IS, IE) seems appropriate for assessing dispositions toward intercultural communication regardless of culture.

Intercultural effectiveness corresponds to the behavioral aspect of ICC. It is understood as the set of verbal and nonverbal communicative skills that show an adaptation of behavior to the situation and context to favor effective communication [42]. Along this same line, Portalla and Chen [40] argue that intercultural effectiveness corresponds to communicative skills and abilities, where verbal and nonverbal behaviors are included, favoring individuals in achieving their communicative objectives in intercultural interaction through adequate and effective acting. Yilmaz et al. [3] point out that intercultural effectiveness enables individuals to provide specific attention and communication to culturally different counterparts effectively. In short, intercultural effectiveness contributes to developing verbal and nonverbal communication skills that help individuals achieve appropriate and effective communication based on accepting differences, considering intercultural dialogue, constant collaboration, and mutual respect.

Concerning the above, teachers in initial teacher training need to develop intercultural effectiveness since, during their educational practice, communication is fundamental in interacting with children, youth, and adults from different cultures. Thus, intercultural effectiveness will contribute to initial teacher training to improve communicative interaction in culturally diverse contexts [16]. In this sense, it will allow future teachers to communicate adequately and effectively within the cultural diversity of the student body and with other people. At the same time, it will contribute to the recognition, understanding, and acceptance of the other, considering the social, cultural, and historical aspects based on intercultural dialogue [8,18,43]. This will generate self-awareness, adaptation, and humility and develop the ability to listen, dialogue, and reflect to innovate and transform teaching practice [44] and, therefore, improve teaching and learning processes. In addition, it will help teachers and students strengthen relationships and the construction of affective ties with members of their own culture and with people from different cultures to act appropriately in contexts marked by cultural diversity [45].

The intercultural effectiveness scale was first developed by Portalla and Chen [40]. It was applied to undergraduate students enrolled in an introductory communication course at a university in the Northeastern United States. Participants in the final stage were 204 students: 74 males and 130 females. These participants were undergraduates enrolled in an introductory communication course. The final version of the IES comprised six factors and 20 items. The results showed that individuals obtained high scores on the IES, evidencing flexible cultural behaviors, ability to distinguish appropriate cultural behaviors, adaptation to specific cultural situations, and ability to maintain appropriate intercultural interaction by demonstrating communication skills. This scale has been used subsequently in different formative contexts to assess the development of this construct, especially among higher education students.

Bates and Rehal [46] used IES to support cross-cultural exploration and development for travel programs conducted by short-term undergraduate students. The IES was applied in an international service-learning program with 13 students and two program leaders in Romania. Participants completed the IES assessment before and after the program. The IES measured the three key dimensions: continuous learning, interpersonal engagement, and robustness. Among the main findings, the IES helped students identify their strengths and weaknesses before travelling abroad. It also helped students identify development, stability, or decline areas based on their intercultural effectiveness. In addition, it allowed program leaders to examine the cross-cultural strengths and weaknesses of each student and the student body as a whole. Results also showed that cultural immersion experiences and time abroad enhanced student learning and development. These findings indicate that the IES helps measure intercultural skills and student intercultural development.

Gungor et al. [2] used the IES and intercultural awareness scale [47] to identify the relationship between intercultural effectiveness and awareness and xenophobia in nursing (*n* = 257) and vocational school of health services (*n* = 341) students. The study reported that female students had significantly higher intercultural effectiveness scores, while male students had higher intercultural awareness scores. Additionally, a negative correlation was found between IES and IAS and the xenophobia scale. The study concludes by determining that the development of intercultural effectiveness and sensitivity in the training of health professionals is required to reduce their xenophobic prejudices.

The IES has been extensively validated in the Turkish context. Arslan et al. [48] validated the IES in a sample of 352 Turkish university students. Their study considered a linguistic validation. The confirmatory factor analysis indicated a good model fit, and the reliability analysis indicated an internal consistency coefficient of 0.83. Yakar and Alpar [49] evaluated the validity and reliability of the ISS and IES in a sample of 204 nurses in Turkey. The study considered a process of linguistic adaptation to Turkish. The confirmatory factor analysis determined that the scale maintained the same number of factors and items as in the original scale of Portalla and Chen [40]. Yilmaz et al. [3] validated the IES in undergraduate nursing students (fourth grade) at two state universities in Izmir, Turkey. A total of 165 students participated in the data collection. Of the students who participated in the study, 83.5% (*n* = 132) were female and 16.5% (*n* = 33) were male. The original 20-item scale was adapted to 15 items and three factors. The results indicate that the opinions of five experts were considered for the linguistic and content validity of the scale. The results show that the Cronbach’s alpha reliability coefficient of the 15-item, three-factor scale was 0.79, suggesting that the scale had sufficient internal consistency.

A correlation has been reported between intercultural effectiveness and intercultural sensitivity. In the study conducted by Portala and Chen [40], a significant correlation was found between IES and ISS with a correlation coefficient of 0.74. This implies that a person with high intercultural sensitivity will better recognize behaviors that are more appropriate during intercultural interaction. On the other hand, Alrasheedi and Almutawa [50] conducted a study in Kuwait that sought to identify the degree of intercultural effectiveness (IE) and intercultural sensitivity (IS) among students of a faculty of education in Kuwait. The sample consisted of 370 students who were randomly selected. The results showed a positive, strong, statistically significant correlation, with values of 0.89 for IES and 0.87 for ISS. On the other hand, Kardas and Yilmaz [51] investigated this correlation in nursing students at Ankara University. The population consisted of 503 undergraduate nursing students. The results show that the mean scores on the ISS scale were 92.56 ± 11.98 and 53.87 ± 6.28 on the IES, and a positive and statistically significant correlation was found between the scores of the two scales.

Despite the studies on the IES in higher education contexts at the international level, these have been insufficient. In the field of initial teacher training in Chile, they are null. In this sense, instruments are necessary to measure intercultural effectiveness reliably and validly in the context of initial teacher training in Chile. This is of concern, given that Chile is a culturally diverse country. In the past five years, Chile has undergone a remarkable multicultural transformation. Immigration, mainly from countries like Venezuela, Haiti, Colombia, and Peru, has enriched the country’s cultural diversity. This wave of migration has impacted various aspects of Chilean society, from education to the labor market. Cities have seen an increase in the diversity of languages, customs, and cultural expressions. This process has brought new challenges regarding integration and public policies as Chilean society adapts to a more plural and globalized identity, reflecting an increasingly multicultural coexistence. Validating the IES in the Chilean context will contribute with inputs in the educational field to help establish appropriate and effective communication that allows promoting recognition and understanding of the other. At the same time, it will provide the basis for further exploration of this area of research using a quantitative methodology and a qualitative methodological approach to deepen and better understand the phenomenon under study. Therefore, the objective of this study was to adapt and validate the intercultural effectiveness scale in a sample of teachers in initial teacher training in Chile. This study posed two hypotheses: (H1) the scores of the intercultural effectiveness scale will meet adequate levels of validity and reliability in the sample of pre-service teachers, and (H2) the scores of the six factors of the IES will present positive and statistically significant correlations with the interaction confidence factor of the ISS scale.

## 2. Materials and Methods

### 2.1. Participants

The research population totaled 1867 teachers in initial training from two Chilean universities. The participants were selected from a multistage, stratified probability sampling process [52] with a reliability of 99.7%, a sampling error of 4.43%, and a variance p = q = 0.5. The sample consisted of 584 students (male = 37.8%; female = 61.6%; other = 0.5%), with an average age of 20.56 years (SD = 3.37) and 21.9% identifying as belonging to an ethnic group.

### 2.2. Instruments

Data collection for this study was conducted through a three-part survey. First, a sociodemographic questionnaire with closed questions was applied: age, sex, commune, name of the establishment where he/she works, dependence, institution where he/she is doing undergraduate studies, and ethnicity.

Second, the intercultural effectiveness scale (IES) [40] was applied. The IES is a self-report instrument that assesses intercultural effectiveness as a behavioral dimension of intercultural communicative competence. The scale has 20 items that are answered on the following scale: 1 = strongly disagree, 5 = strongly agree. The IES has a structure of six correlated factors called behavioral flexibility (4 items, e.g., “2. I am afraid to express myself when interacting with people from different cultures.”), interaction relaxation (5 items, e.g., “1. I find it easy to talk with people from different cultures.”), interactant respect (3 items, e.g., “15. I always show respect for my culturally different counterparts during our interaction.”), message skills (3 items, e.g., “6. I have problems with grammar when interacting with people from different cultures.”), identity maintenance (3 items, e.g., “8. I find it is difficult to feel my culturally different counterparts are similar to me.”), and interaction management (2 items, e.g., 7. “I am able to answer questions effectively when interacting with people from different cultures.”).

Third, we applied the version adapted for Chile [18] of the intercultural sensitivity scale (ISS) [39], which evaluates intercultural sensitivity as an affective dimension of intercultural competence. It is a 5-point Likert-type scale (1 = strongly disagree; 2 = disagree; 3 = neutral; 4 = agree; 5 = strongly agree), composed of 24 items distributed in the five factors named interaction engagement (7 items, e.g., “1. I enjoy interacting with people from different cultures.”), respect for cultural difference (6 items, e.g., “2. I think people from other cultures are narrow-minded.”), interaction confidence (5 items, e.g., “4. I find it very hard to talk in front of people from different cultures.”), interaction enjoyment (3 items, e.g., “9. I get upset easily when interacting with people from different cultures.”), and the interaction attentiveness factor (3 items, e.g., “14. I am very observant when interacting with people from different cultures.”). Regarding validity and reliability indicators, the scale has reported acceptable indicators for five correlated factors [53,54,55]. Regarding Spanish adaptations, Sanhueza [56] reported a two-factor correlated structure and Micó-Cebrian and Cava [57] a unidimensional structure.

### 2.3. Procedure

The IES underwent a linguistic adaptation process [58]. The research team used forward–backward–forward translation techniques to verify semantic consistency. Then, the IES was reviewed by monolinguals in each of the languages. Finally, the research team selected the final version of the adapted scale.

Contact was made with the university program directors, who signed a letter of agreement for the fieldwork. Subsequently, informed consent was applied, following the protocol authorized by the Universidad de La Frontera (Nº090/23), safeguarding the project’s ethical principles. Finally, the students responded voluntarily and anonymously to the measurement scales.

### 2.4. Data Analysis

In order to reach the objective of this research, descriptive analyses and univariate normality assessments were performed for the 20 items of the scale using SPSS v.25 software. Then, a confirmatory factor analysis (CFA) was performed using MPLUS v.8.1 software [59], using the maximum likelihood estimation method with robust standard errors (MLR). To evaluate the model, the following goodness-of-fit indices were analyzed: comparative fit index (CFI) and Tucker–Lewis index (TLI). A value greater than or equal to 0.90 was considered a good fit for both indexes. In addition, the root mean square error of approximation (RMSEA) was used. For this index, a value less than or equal to 0.08 was considered a reasonable fit. Finally, the standardized root mean square residual (SRMR), with a value less than or equal to 0.07, was considered a reasonable fit [60]. McDonald’s ω and Cronbach’s α coefficients were used to estimate reliability [61].

## 3. Results

### 3.1. Descriptive Analysis

As can be seen in Table 1, the highest mean corresponds to item 20, “I always show respect for the opinions of my culturally different counterparts during our interaction.” (mean = 4.29; SD = 0.842) and the lowest mean was for item 12 “I often miss parts of what is going on when interacting with people from different cultures.” (mean = 2.92; SD = 0.863). In addition, the univariate normality of the scale items was assessed, as shown in Table 1. All items failed to meet the assumption of univariate normality.

### 3.2. Evidence of Validity

A confirmatory factor analysis was applied to evaluate the factor structure of the IES (Figure 1). The first model tested was unidimensional, and the results were unsatisfactory (MLR-*χ*^2^ (df = 152) = 833.051; *p* < 0.01; RMSEA = 0.083 [C.I. = 0.078–0.089]; CFI = 0.840; TLI = 0.820; SRMR = 0.067). Subsequently, we proceeded to evaluate the original factor structure. The results showed a good fit to the solution of six correlated factors (MLR-*χ*^2^ (df = 155) = 488.297; *p* < 00.1; RMSEA = 0.058 [C.I. = 0.052–0.063]; CFI = 0.923; TLI = 0.906), however, the factor loading of item 18 was insufficient (l = −0.84; *p* = 0.91). Finally, the model was re-estimated with 19 items, obtaining satisfactory results (MLR-*χ*^2^ (df = 137) = 436.859; *p* < 0.01; RMSEA = 0.058 [C.I. = 0.052–0.064]; CFI = 0.930; TLI = 0.912; SRMR = 0.049).

Concerning the evidence of convergent validity, Pearson’s r correlations were estimated between the IES’s six factors and the interaction confidence factor of the ISS scale. The results showed positive correlations, statistically significant and in the expected theoretical sense: behavioral flexibility (r = 0. 336: *p* < 0.01), interaction relaxation (r = 0.661: *p* < 0.01), interactant respect (r = 0.387: *p* < 0.01), message skills (r = 0.204: *p* < 0.01), identity maintenance (r = 0.319: *p* < 0.01), and interaction management (r = 0.478: *p* < 0.01).

### 3.3. Evidence of Reliability

Table 2 presents the results of the IES’s reliability analysis. As can be seen in Table 2, the interactant respect factor presents the highest reliability results. In contrast, the identity maintenance factor presents the lowest reliability values. However, all values present adequate reliability levels.

Although reliability levels vary among the factors assessed, all factors reach adequate levels of reliability. This indicates that the IES is a consistent and adequate scale for measuring the intercultural effectiveness construct, which reinforces the robustness of the instrument in the context of the present research.

## 4. Discussion

The objective of this study was to adapt and validate the intercultural effectiveness scale (IES) in a sample of teachers in initial training in Chile. The results demonstrate the achievement of the two hypotheses initially proposed.

The first hypothesis, which stated that the intercultural effectiveness scale would meet adequate levels of validity in the sample of teachers in initial training, was achieved since the best statistical model was that of six correlated factors consistent with the original theoretical model, in addition to adequate levels of reliability. These results are consistent with previous studies [3,40,47,49], which established that this factor structure is the most suitable for this instrument.

For the second hypothesis, which indicated that the scores of the six factors of the IES would present positive and statistically significant correlations with the interaction confidence factor of the ISS scale, the results evidenced a total achievement, given that all the factors of the IES correlated significantly with the interaction confidence factor. These results are consistent with previous findings, which have correlated both constructs (intercultural sensitivity and intercultural effectiveness). Particularly in the study by Kardas and Yilmaz [50], a positive, moderate, and statistically significant correlation was found between the results of both scales. This study also found that females had a higher SI and EI than males and that students who did not speak English had a lower SI and EI than those who could understand and speak English.

Regarding the final number of items, the factor load of item 18 was insufficient, so the adaptation to the Chilean context considers only the 19 items with an appropriate factor load. In this sense, it would be necessary to determine whether the item’s linguistic adaptation requires revision, given that in other studies in which linguistic adaptations have been made to other languages, the item has obtained adequate values and has been maintained in the final adapted versions [47,49].

Although the identity maintenance factor presented the lowest reliability values, all the values present adequate reliability levels, which shows that the scale has sufficient internal consistency, given that the reliability coefficient of the original scale of Chen and Starosta [41] was 0.85. Indeed, the scale’s stability over time remained strong.

Some limitations of the present study correspond to the study’s cross-sectional nature, which prevents the analysis of trends in the sample studied. In addition, the sample size was limited and represented only one area of professionals in training (teaching). These limitations suggest new studies in which a larger and more heterogeneous sample could be considered to validate this scale in a more representative population sample. Additionally, the projections of this study would involve conducting longitudinal studies with a more heterogeneous sample of teachers in initial training from state and private universities in Chile. Likewise, it is recommended to apply the intercultural effectiveness scale to university teachers who train future teachers, given that the literature indicates that, in order to develop interculturally competent teachers, it is required that their teachers (teacher educators) foster the development of this competence during initial training. Without formative experiences for intercultural reflection, novice teachers fail to address deep cultural differences among their students [62]. Applying this scale to students attending graduate programs in intercultural contexts is also suggested. Finally, validating the IES in intercultural school contexts is recommended, considering an appropriate linguistic adaptation to the population of preadolescents and adolescents and carrying out a longitudinal study in this context since there is no literature on the subject. Intercultural education has been addressed in Chile at the school level for the recognition and inclusion of native peoples. However, little has been studied on intercultural communication in the context of global citizenship education, which is adequately inserted in the growing multicultural scenarios that are currently present in the local context [63].

## 5. Conclusions

The results show that the linguistic adaptation to Spanish of the IES met adequate validity and reliability levels, maintaining the original scale’s five factors [40]. Likewise, the correlation between the ISS and the scores of the six factors of the IES presented positive and statistically significant correlations, indicating that the instruments account for the constructs being assessed. Having validated instruments for the Chilean context will allow understanding of the state of development in intercultural communicative competence of teachers in initial training, which will contribute to the generation of information necessary for the training of teachers with flexible and adequate communicative behaviors to favor and maintain appropriate and effective intercultural communication in intercultural educational contexts.

## Figures and Tables

**Figure 1 behavsci-14-00864-f001:**
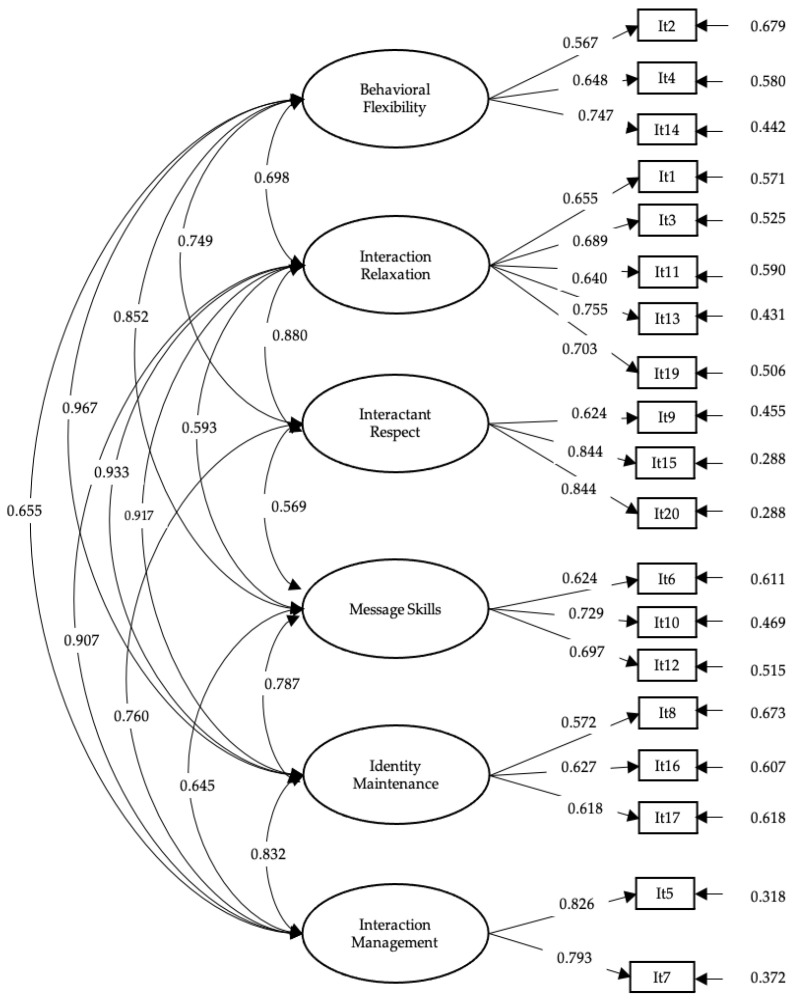
Factor structure for the intercultural effectiveness scale.

**Table 1 behavsci-14-00864-t001:** Descriptive statistics.

Items	Mean	Standard Deviation	Asymmetry	Kurtosis	K-S Test
It1	3.54	0.971	−0.237	−0.19	0.216 **
It2	3.39	1.049	−0.158	−0.612	0.186 **
It3	3.67	0.956	−0.605	0.392	0.234 **
It4	3.42	1.071	−0.25	−0.458	0.190 **
It5	3.62	0.865	−0.347	0.08	0.242 **
It6	2.96	0.97	0.053	−0.196	0.223 **
It7	3.47	0.845	−0.172	0.115	0.234 **
It8	3.48	1.106	−0.288	−0.575	0.179 **
It9	3.81	0.952	−0.736	0.446	0.258 **
It10	3.04	0.851	0.14	0.294	0.273 **
It11	3.15	0.999	0.016	−0.32	0.224 **
It12	2.92	0.863	0.108	−0.042	0.236 **
It13	3.48	0.915	−0.276	−0.01	0.208 **
It14	3.59	1.008	−0.536	−0.027	0.232 **
It15	4.24	0.851	−1.092	1.343	0.272 **
It16	3.46	0.949	−0.237	−0.196	0.203 **
It17	3.31	0.757	0.309	0.879	0.341 **
It18	4.15	0.887	−0.8	0.154	0.260 **
It19	3.41	0.775	0.061	0.775	0.307 **
It20	4.29	0.842	−1.061	0.855	0.306 **

Note: K-S Test = Kolmogorov–Smirnov Test, ** *p* < 0.01.

**Table 2 behavsci-14-00864-t002:** Scale reliability statistics.

Factor	McDonald’s ω	Cronbach’s α
Behavioral Flexibility	0.653	0.636
Interaction Relaxation	0.787	0.786
Interactant Respect	0.820	0.815
Message Skills	0.692	0.691
Identity Maintenance	0.593	0.573
Interaction Management	0.761	0.761

## Data Availability

The dataset for this study is available from the corresponding author upon reasonable request due to ethical restrictions.

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
