# Peer review of "Adaptation and Validation of the Intercultural Effectiveness Scale in a Sample of Initial Teacher Training Students in Chile"

_behavsci, 2024, doi:10.3390/bs14100864_

Round 1
Reviewer 1 Report
Comments and Suggestions for Authors
The topic of the article and the problem of the research presented in it is really relevant and actual and in future even more. The theoretical backgroud also presented qualitatively as well as the methodological part of the research. The main goal is reached and the results are presented visualy but it would be necesarry as well as even more useful to add some comments and analysis of the data presented in the tables, for example, lines 272, 297 and etc. At least because it is not recommended to finish the chapter with table or figure.
Comments on the Quality of English Language-
Reviewer 2 Report
Comments and Suggestions for Authors
1. The manuscript is well framed and interesting to read. One issue I suggest to be included in the introduction is the rationale for adaptation and validation in the Chilean context. It is good if the authors explain why adaptation and validation of the tool is necessary in the Chilean context.
2. It is not clear why the authors considered the Interaction Confidence factor of the Intercultural Sensitivity Scale (ISS) to examine the convergent validity of the adapted IES factors. The ISS has five factors. However, the authors only considered the Interaction Confidence Scale to check convergent validation. Why is it chosen? It’s good if the authors provide explanation about it.
3. Some minor corrections need to be made. For example, between lines 78 and 93, Inter-Cultural Competence is abbreviated as CCI when it should actually be abbreviated as ICC. The p value in line 281 is given as p<00.1. I assume that p<0.01 was intended.
Reviewer 3 Report
Comments and Suggestions for Authors
First of all, thank you very much for the opportunity to review such an interesting manuscript. Below are some recommendations for the authors:
Abstract
- I recommend placing the abbreviation (ISS) in parentheses, following the trend for defining acronyms on line 14.
Introduction
- First of all, the format of the references is not appropriate for the journal. For example, the first citation in the text should appear as [1-3] and on line 28 as [5,6]. Please review this issue throughout the manuscript.
- I recommend not using “&” in the text, replace it with “and”.
- The paragraph that begins on line 121 is a section that should appear in the “Material and Method”. It is correct to communicate how the questionnaire was validated but there is no need to reference its psychometric properties in the introduction.
- Likewise, the paragraphs that follow are full of information at the level of results, which could perfectly be placed in the Discussion. Please, if you want to keep this information in the introduction section, please summarize it as much as possible.
Materials and Methods
- Line 256: “For model evaluation, the following goodness-of-fit indices were analyzed: comparative fit index (CFI) and Tucker-Lewis Index (TLI). For both indices, a value greater than or equal to 0.90, root mean square error of approximation (RMSEA), and standardized root mean square residual (SRMR) were considered, and a value less than or equal to 0.07 was considered a reasonable fit [60].”
There is no cohesion at the wording level between the first and second sentences, please correct this aspect.
Results
- I do not understand the normality analysis. It is understood that since these are ordinal Likert-type responses, the data do not meet the normality hypothesis.
- I have serious doubts about the validity of the factor model… By eliminating item 18, the last factor is left with only 2 items, violating the recommendations of various authors in the field [1,2]. In this particular case, I would recommend that you carry out a prior Exploratory Factor Analysis.
1. Raubenheimer, J. An Item Selection Procedure to Maximise Scale Reliability and Validity. SA Journal of Industrial Psychology 2004, 30, doi:10.4102/sajip.v30i4.168.
2. MacCallum, R.C.; Widaman, K.F.; Zhang, S.; Hong, S. Sample Size in Factor Analysis. Psychological Methods 1999, 4, 84–99, doi:10.1037/1082-989X.4.1.84.
Discussion and Conclusions
- The information provided is correct
